# Novel Conjugated Polymers Containing 3-(2-Octyldodecyl)thieno[3,2-*b*]thiophene as a π-Bridge for Organic Photovoltaic Applications

**DOI:** 10.3390/polym12092121

**Published:** 2020-09-17

**Authors:** Jong-Woon Ha, Jong Baek Park, Hea Jung Park, Do-Hoon Hwang

**Affiliations:** Department of Chemistry and Chemistry Institute for Functional Materials, Pusan National University, Busan 46241, Korea; newhju@naver.com (J.-W.H.); magician_p@naver.com (J.B.P.); heajungp@pusan.ac.kr (H.J.P.)

**Keywords:** organic photovoltaic, 3-(2-octyldodecyl)thieno[3,2-*b*]thiophen, fullerene, high crystallinity

## Abstract

3-(2-Octyldodecyl)thieno[3,2-*b*]thiophen was successfully synthesized as a new π-bridge with a long branched side alkyl chain. Two donor-π-bridge-acceptor type copolymers were designed and synthesized by combining this π-bridge structure, a fluorinated benzothiadiazole acceptor unit, and a thiophene or thienothiophene donor unit, (**PT-ODTTBT** or **PTT-ODTTBT** respectively) through Stille polymerization. Inverted OPV devices with a structure of ITO/ZnO/polymer:PC_71_BM/MoO_3_/Ag were fabricated by spin-coating in ambient atmosphere or N_2_ within a glovebox to evaluate the photovoltaic performance of the synthesized polymers (effective active area: 0.09 cm^2^). The **PTT-ODTTBT**:PC_71_BM-based structure exhibited the highest organic photovoltaic (OPV) device performance, with a maximum power conversion efficiency (PCE) of 7.05 (6.88 ± 0.12)%, a high short-circuit current (*J*_sc_) of 13.96 mA/cm^2^, and a fill factor (*FF*) of 66.94 (66.47 ± 0.63)%; whereas the **PT-ODTTBT**:PC_71_BM-based device achieved overall lower device performance. According to GIWAXS analysis, both neat and blend films of **PTT-ODTTBT** exhibited well-organized lamellar stacking, leading to a higher charge carrier mobility than that of **PT-ODTTBT**. Compared to **PT-ODTTBT** containing a thiophene donor unit, **PTT-ODTTBT** containing a thienothiophene donor unit exhibited higher crystallinity, preferential face-on orientation, and a bicontinuous interpenetrating network in the film, which are responsible for the improved OPV performance in terms of high *J*_sc_, *FF*, and PCE.

## 1. Introduction

Bulk heterojunction organic photovoltaics (OPVs) have received a great deal of attention owing to their unique advantages including low cost, mechanical flexibility, and ease processability [1,2,3,4,5]. Recently, their photovoltaic performance has rapidly advanced through the development of state-of-the-art electron donor/acceptor or interfacial materials, as well as optimization of the OPV device fabrication process via adjusting additives, solvents, and thermal treatment. They showed enhanced power conversion efficiencies (PCEs) of over 15% and 17% for single junction devices and double junction devices, respectively [6,7,8,9,10]. However, the problems such as the difficulty in controlling the morphology of the active layer, the sensitive device efficiency according to film thickness, and the polymer solubility in organic solvents have still remained for real application through roll-to-roll and ink-jet printing processes. It has been studied to find effective methods to solve the above problems through molecular engineering. [11,12,13,14,15]. The active layer of an OPV is composed of an interpenetrating network formed by blending electron-donor and electron-acceptor materials [16,17]. In particular, π-conjugated polymers with a donor(D)–π–bridge–acceptor(A) architecture are commonly used as an electron donor. By changing the molecular structures of the D and A moieties, the properties of the conjugated polymer such as the energy levels of the highest occupied molecular orbital (HOMO) and the lowest unoccupied molecular orbital (LUMO), absorption range, charge carrier mobility, and morphology can be adjusted [18,19,20]. Moreover, the π-bridge unit as a component of the polymer backbone crucially affects the molecular structure and electronic properties of the polymer, and consequently impacts the physical and optoelectronic properties of the D–π–A type conjugated polymer. Therefore, choosing the proper π-bridge could be an important strategy for enhancing the OPV performance. Among the D–π–A type polymer donors, benzothiadiazole based polymer donors have been reported with excellent photovoltaic properties. Notably, H. Yan et al. reported a promising polymer donor (PffBT4T-2OD) consisting of benzothiadiazole as an electron accepting building block. The fabricated fullerene-based OPV showed significantly high PCE of 11% with excellent solubility arising from the incorporation of the appropriate alkyl-substituted π-bridge onto the polymer backbone [20].

The fused heterocyclic ring of thieno[3,2-*b*]thiophene (TT) is attractive as an electron-donating building block or π-bridge in D–π–A type conjugated polymers, because of its excellent electron-donating ability and good planarity [21,22,23,24,25]. In particular, polymer donors containing TT exhibited a well-organized crystal domain and high charge carrier mobility and can be used in the development of efficient OPVs. We have previously reported several new D–π–A type polymers consisting of linear alkyl-substituted TT as a π-bridge (PBDT–TPD, PBDT–ttTPD, PBDTT–TPD, and PBDTT–ttTPD) that showed excellent OPV device performance in single/tandem solar cells owing to their well-ordered orientation and enhanced hole mobility [22]. Additionally, Wang et al. reported that the polymer donor P(BDT-TT-BT), which contained TT as the π-bridge instead of thiophene (T), has a broad absorption spectrum and increased hole mobility because of the extended π-conjugation and enhanced crystallinity [24]. However, incorporation of TT limits the solubility of polymer in common organic solvents, since the rigid planar structure also resulted in strong *π*–*π* stacking of the polymer. To address these issues, a linear alkyl side chain was substituted onto TT to improve the solubility of the polymer in organic solvents [22,24] but sufficient solubility is still required to facilitate the subsequent device fabrication. Conventional synthetic approaches to introduce a longer branched alkyl side chain onto TT using a palladium or nickel complex as a catalyst afforded only trace chemical yield. Moreover, the procedure was inconvenient as the organometallic reagents need to be pre-generated by metal insertion (Appendix A).

In this study, we successfully synthesized 3-(2-octyldodecyl)thieno[3,2-*b*]thiophen by cobalt (II)-catalyzed reductive alkylation as a new π-bridge to improve the solubility and crystallinity of conjugated polymers incorporating TT [26]. Two desired polymers consisting of 5,6-difluorobenzo[*c*][1,2,5]thiadiazole as an electron accepting unit and T or TT as an electron donating unit were design and synthesized, namely, poly(5,6-difluoro-4-(6-(2-octyldodecyl)-5-(thiophen-2-yl)thieno[3,2-*b*]thiophen-2-yl)-7-(6-(2-octyldodecyl)thieno[3,2-*b*]thiophen-2-yl)benzo[*c*][1,2,5]thiadiazole) (**PT-ODTTBT**) and poly(5,6-difluoro-4-(3-(2-octyldodecyl)-[2,2′-bithieno[3,2-*b*]thiophen]-5-yl)-7-(6-(2-octyldodecyl)thieno[3,2-*b*]thiophen-2-yl)benzo[*c*][1,2,5]thiadiazole) (**PTT-ODTTBT**). Their photovoltaic properties were investigated and compared.

## 2. Materials and Methods

### 2.1. Materials

All reagents were purchased from Sigma Aldrich (St. Louis, MO, USA) and used without further purification. 3-Bromothieno[3,2-*b*]thiophene, 2,5-bis(trimethylstannyl)thiophene (M1), and 2,5-bis(trimethylstannyl)thieno[3,2-*b*]thiophene (M2) were synthesized according to previously reported methods [27,28,29].

### 2.2. Monomer Syntheses

#### 2.2.1. Synthesis of 3-(2-octyldodecyl)thieno[3,2-b]thiophene (**1**)

3-Bromothieno[3,2-*b*]thiophene (10.0 g, 45.6 mmol) and 9-(iodomethyl)nonadecane (22.4 g, 54.8 mmol) were added to a solution of tri(*o*-tolyl)phosphine (2.8 g, 9.1 mmol), cobalt (II) bromide (2.0 g, 9.1 mmol), and manganese powder (10.0 g, 182.5 mmol) in *N*,*N*-dimethylacetamide (80 mL) and pyridine (20 mL) under N_2_. Five drops of trifluoroacetic acid was slowly added, and then the reaction mixture was heated to 70 °C for 24 h. After the reaction mixture was cooled to room temperature, the precipitates were passed through celite and washed with ethyl acetate. Aqueous ammonium chloride was added to the filtrate, and the mixture was extracted with ethyl acetate three times. The collected organic layer was washed with brine, dried using anhydrous magnesium sulfate (MgSO_4_), and concentrated in vacuo. The crude product was purified by silica column chromatography using hexane as an eluent to yield the desired product **1** (10.1 g, 52.1%) as a colorless oil. ^1^H NMR (CDCl_3_, 300 MHz): δ (ppm) 7.35 (d, *J* = 5.1 Hz, 1H), 7.24 (d, *J* = 5.1 Hz, 2H), 6.95 (s, 1H), 2.65 (d, *J* = 6.9 Hz, 2H), 1.83 (m, 1H), 1.23 (m, 32H), 0.87 (m, 6H).

#### 2.2.2. Synthesis of tributyl(6-(2-octyldodecyl)thieno[3,2-b]thiophen-2-yl)stannane (**2**)

*n*-Butyllithium (7.1 mL, 14.3 mmol, 2.0 M in cyclohexane) was added dropwise to a solution of **1** (6.0 g, 14.3 mmol) in anhydrous tetrahydrofuran at −78 °C. After stirring at −78 °C for 15 min, tributyltin chloride (5.1 g, 15.7 mmol) was added. Subsequently, the reaction mixture was gradually warmed to room temperature and further stirred for 1 h. The reaction mixture was quenched with water and extracted with dichloromethane three times. The collected organic layer was dried over anhydrous MgSO_4_ and then concentrated in vacuo. The crude product was used without purification for the next step. ^1^H NMR (CDCl_3_, 300 MHz): δ (ppm) 7.23 (s, 1H), 6.91 (s, 1H), 2.64 (s, *J* = 6.3 Hz, 2H), 1.84 (m, 1H), 1.52 (m, 50H), 0.92 (m, 15H).

#### 2.2.3. Synthesis of 5,6-difluoro-4,7-bis(6-(2-octyldodecyl)thieno[3,2-b]thiophen-2-yl)benzo[c][1,2,5]thiadiazole (**3**)

4,7-Dibromo-5,6-difluorobenzo[c][1,2,5]thiadiazole (1.0 g, 3.0 mmol) and dichlorobis(triphenylphosphine)palladium(II) (63.8 mg, 0.1 mmol) were dissolved in anhydrous *N*,*N*-dimethylformamide. After stirring at 140 °C for 30 min, **2** was added. After stirring overnight at the same temperature, the reaction mixture was extracted with dichloromethane and washed with brine. The collected organic layer was dried over anhydrous MgSO_4_ and concentrated under reduced pressure. The residue was purified by silica column chromatography with hexane as the eluent to obtain a red oil (2.4 g, 78.4%). ^1^H NMR (CDCl_3_, 300 MHz): δ (ppm) 8.55 (s, 2H), 7.08 (s, 2H), 2.74 (d, *J* = 6.9 Hz, 4H), 1.92 (m, 2H), 1.29 (m, 64H), 0.94 (m, 12H).

#### 2.2.4. Synthesis of 4,7-bis(5-bromo-6-(2-octyldodecyl)thieno[3,2-b]thiophen-2-yl)-5,6-difluorobenzo[c][1,2,5]thiadiazole (**ODTTBT**)

To a solution of **3** (1.0 g, 1.0 mmol) in *N*,*N*-dimethylformamide, *N*-bromosuccinimide (NBS) (0.35 g, 2.0 mmol) was added and stirred at room temperature overnight in the dark. Water was then poured into the reaction mixture and extracted with dichloromethane, and the organic layer was dried over anhydrous MgSO_4_. After removal of the solvent under reduced pressure, the residue was recrystallized from dichloromethane/methanol to afford the product as a purple solid (0.76 g, 65.7%). ^1^H NMR (CDCl_3_, 300 MHz): δ (ppm) 8.46 (s, 2H), 2.72 (d, *J* = 7.2 Hz, 4H), 2.01 (m, 2H), 1.24 (m, 64H), 0.85 (m, 12H).

### 2.3. Polymerization Procedure

A mixture of the distannylated monomer (1 equivalent), **ODTTBT** (1 equivalent), and tetrakis(triphenylphosphine)palladium(0) (Pd(PPh_3_)_4_, 0.03 equivalent) was dissolved in anhydrous toluene (5 mL) and *N*,*N*-dimethylformamide (1 mL). The reaction mixture was stirred at 100 °C for 16 h, and then 2-tributhylstannylthiophene (0.2 mL) and 2-bromothiophene (0.2 mL) were added as end-cappers. After another 2 h, the reaction mixture was poured into methanol (200 mL), and the precipitate was collected by filtration and purified by Soxhlet extraction using methanol, acetone, and hexane. The polymers were obtained by reprecipitation of the chloroform solution in methanol.

#### 2.3.1. **PT-ODTTBT**

**ODTTBT** (0.25 g, 0.20 mmol), M1 (87.7 mg, 0.20 mmol), and Pd(PPh_3_)_4_ (7.4 mg) were used to synthesize **PT-ODTTBT** via the previously described method (170 mg, 78%). GPC: *M*_n_ = 82.3 Kda, *M*_w_ = 103 Kda, *PDIs* = 1.38, *T*_d_ = 388 °C.

#### 2.3.2. **PTT-ODTTBT**

**ODTTBT** (0.37 g, 0.30 mmol), M2 (0.15 mg, 0.30 mmol), and Pd(PPh_3_)_4_ (11.0 mg) were used to synthesize **PTT-ODTTBT** via the previously described method (310 mg, 90%). GPC: *M*_n_ = 95.4 kDa, *M*_w_ = 119 kDa, *PDIs* = 1.54, *T*_d_ = 384 °C.

## 3. Results and Discussion

### 3.1. Synthesis and Characterization of Polymers

The key precursor **1** for the synthesis of **ODTTBT** was obtained by the reaction of 3-bromothieno[3,2-*b*]thiophene and 2-iodooctyldodecan using cobalt (II)-catalyzed alkylation to afford a colorless liquid. Compound **1** was subjected to stannylation at the α position to obtain **2**, which was used to synthesize **3** through the Stille cross-coupling reaction with 4,7-dibromo-5,6-difluorobenzo[*c*][1,2,5]thiadiazole. Finally, **3** was reacted with *N*-bromosuccinimide (NBS) to yield the final brominated monomer **ODTTBT**. The synthesized ODTTBT was reacted with either thiophene (M1) or thienothiophene distannylated monomer (M2) via Stille polymerization to give **PT-ODTTBT** or **PTT-ODTTBT**, respectively, as shown in Scheme 1. The detailed synthetic procedures of **ODTTBT** and the polymers are provided in the Experimental Section. The synthesized polymers were found to be soluble in chloroform, chlorobenzene (CB), and *o*-dichlorobenzene. The number-average molecular weight (*M*_n_)/polydispersity index (*PDI*) was 82.3 kDa/1.26 and 95.1 kDa/1.25 for **PT-ODTTBT** and **PTT-ODTTBT**, respectively, as measured by gel-permeation chromatography (GPC) using chloroform as an eluent and polystyrene as a reference. (Appendix A) Thermal properties of the synthesized polymers were characterized by thermogravimetric analysis (TGA) and differential scanning calorimetry (DSC). All the polymers exhibited good thermal stability, had thermal decomposition temperatures (*T*_d_, 5% weight loss temperature) above 380 °C, and showed no signal for endo- and exo-thermic processes in the heat flow when measured using DSC, indicating their suitability for the OPV fabrication (Appendix A).

Density functional theory (DFT) calculation at the B3LYP/6-31G(d) level was conducted to obtain the optimized structures and frontier molecular orbitals for **PT-ODTTBT** and **PTT-ODTTBT**, using the dimeric architectures for each polymer to simplify the computation. The results indicated that the two polymers have similar geometric structures and frontier molecular orbital distributions (Appendix A). The HOMO was delocalized on the entire backbone, whereas the LUMO was localized at the BT unit, which would generate efficient charge transfer between the electron-donating and electron-accepting building blocks. The calculated HOMO/LUMO energy levels were −5.03/−2.88 eV and −5.01/−2.91 eV for **PT-ODTTBT** and **PTT-ODTTBT**, respectively. The HOMO of **PTT-ODTTBT** was higher in energy than that of **PT-ODTTBT**, because TT on the polymer backbone has a stronger electron donating capability than T.

### 3.2. Optical and Electrochemical Properties

Figure 1a shows the absorption spectra of **PT-ODTTBT** and **PTT-ODTTBT** in the CB solution and in the thin film states. The copolymers displayed broad absorption from 400 to 750 nm with a strong absorption band in the longer wavelength region, owing to the intramolecular charge transfer between the electron-donating and electron-accepting building blocks. **PT-ODTTBT** showed slightly red-shifted absorption at 681 nm in the film state compared with that in the solution because of enhanced molecular aggregation in the solid state. In contrast, **PTT-ODTTBT** exhibited similar absorption maxima in the two states (688 nm), indicating that this polymer undergoes strong molecular aggregation even in the solution state owing to favorable *π*–*π* stacking upon introducing the TT block on the polymer backbone. The optical bandgap (*E*_g_^opt^) was estimated to be 1.65 eV for **PT-ODTTBT** and 1.69 eV for **PTT-ODTTBT** from the corresponding absorption edge in the film state.

The electrochemical properties of **PT-ODTTBT** and **PTT-ODTTBT** were investigated by cyclic voltammetry (Appendix A). From their first oxidation onset potentials, the HOMO energy levels of **PT-ODTTBT** and **PTT-ODTTBT** were estimated to be −5.48 and −5.26 eV, respectively. **PTT-ODTTBT** possessed a higher HOMO energy than **PT-ODTTBT**. The LUMO energies of the synthesized polymers, estimated by combining their *E*_g_^opt^ and HOMO energy levels, were −3.83 and −3.57 eV for **PT-ODTTBT** and **PTT-ODTTBT**, respectively. These experimentally measured energy levels for the frontier molecular orbitals show similar trends to the DFT calculation results. The optical and electrochemical properties of the synthesized polymers are summarized in Table 1. Figure 1b is a schematic energy diagram of the synthesized donor polymers and PC_71_BM acceptor. It shows an energy cascade that can facilitate efficient charge dissociation and transportation to the electrodes in the OPV device.

### 3.3. Photovoltaic Characteristics

Inverted OPV devices with a structure of indium-tin oxide (ITO)/ZnO/polymer:PC_71_BM/MoO_3_/Ag were fabricated to evaluate the photovoltaic performance of **PT-ODTTBT** and **PTT-ODTTBT**. The OPV devices were systematically optimized by controlling the donor/acceptor blending ratio, additive solvent, and thickness of the photoactive layer (Appendix A). The current density-voltage (*J*-*V*) and the corresponding external quantum efficiency (EQE) curves are shown in Figure 2. The **PTT-ODTTBT**:PC_71_BM blend exhibited a slightly lower *V*_oc_ than **PT-ODTTBT**:PC_71_BM (0.75 vs. 0.77 eV) because of the relatively higher HOMO energy of **PTT-ODTTBT**. (Table 2) However, the **PTT-ODTTBT**:PC_71_BM blend showed higher *J*_sc_ (13.13 mA/cm^2^) and *FF* (67.88%) than the PT-ODTTBT:PC_71_BM blend under the same conditions. This might be attributed to the improved crystallinity and hole mobility of the polymer film upon incorporation of TT on the polymer backbone. 1,8-Diiodooctane (DIO) was selected as an additive solvent to control the morphology of the photoactive layer, and its addition enhanced *J*_sc_ of both **PT-ODTTBT**- and **PTT-ODTTBT**-based OPVs. The **PTT-ODTTBT**:PC_71_BM blend exhibited an improved PCE of 7.05% with *V*_oc_ of 0.75 V, *J*_sc_ of 13.96 mA/cm^2^, and *FF* of 66.94%, whereas the **PT-ODTTBT**:PC_71_BM blend exhibited overall lower device performance. To further study this, hole-only devices consisting of ITO/PEDOT:PSS/photoactive layer/MoO_3_/Ag were fabricated using the blend films to measure the space-charge limited current (SCLC), as depicted in Figure 3. The measured hole mobility of **PTT-****ODTTBT**:PC_71_BM (7.08 × 10^−3^ cm^2^ v^−1^ s^−1^) was one order of magnitude higher than that of **PT-ODTTBT**:PC_71_BM (2.55 × 10^−4^ cm^2^ v^−1^ s^−1^). This reveals that the incorporation of TT as an electron-donating building block on the polymer backbone can improve the hole mobility. The corresponding EQE spectra of the **PT-ODTTBT**- and **PTT-ODTTBT**-based OPV devices are provided in Figure 2b. The **PTT-ODTTBT**:PC_71_BM device without DIO exhibited a response range from 300 to 750 nm, and the EQE value exceeded 50% from 400 to 700 nm with a maximum EQE of 57% at 540 nm. After adding DIO, the EQE curves of **PTT-****ODTTBT**:PC_71_BM increased by approximately 60% compared to the device without DIO.

Grazing incidence wide-angle X-ray scattering (GIWAXS) was used to investigate the effect of crystallinity and molecular orientation of the neat polymer and blend films on the device performance. Figure 4 shows the GIWAXS images of the neat polymer and blended films, as well as the corresponding line-cut profiles in the out-of-plane (OOP) and in-plane (IP) directions. Neat films of **PT-ODTTBT** and **PTT-ODTTBT** exhibited distinct (100) diffraction peaks in both OOP and IP directions. Interestingly, a well-ordered lamellar stacking reflection peak from (100) to (300) along the OOP direction was observed in neat **PTT-ODTTBT**. This demonstrates that introduction of TT on the polymer backbone enhanced the molecular crystallinity compared to that of T [30]. The neat **PT-ODTTBT** film exhibited favorable (010) diffraction of *π*–*π* stacking along the OOP direction, indicating a preferential face-on orientation. In contrast, the neat **PTT-ODTTBT** film showed both edge-on and face-on orientations. After blending **PT-ODTTBT** with PC_71_BM, the (010) peak disappeared, indicating a decrease in the crystallinity of the blend film. In contrast, **PTT-ODTTBT** maintained its crystallinity in the blend film by showing strong lamellar stacking diffraction (Figure 4a). It reveals that **PTT-ODTTBT** can maintain higher crystallinity in the blended film than **PT-ODTTBT**, resulting in increased *J*_sc_ and charge carrier mobility. An azimuthal scan corresponding to the lamellar stacking (100) diffraction was conducted to further understand the correlation between molecular orientation and device efficiency [31]. According to Figure 5 and Appendix A, the blend films of **PT-ODTTBT** and **PTT-ODTTBT** displayed similar trends: the face-on ratio gradually increased upon adding DIO to the polymer:PC_71_BM blend films (Figure 4b). The integrated ratios in the face-on region for **PTT-****ODTTBT**:PC_71_BM were 25.5% without DIO and 37.7% with DIO, whereas lower integrated ratios (12.1% and 20.8%, respectively) were observed in **PT-ODTTBT**:PC_71_BM. Compared to PT-ODTTBT:PC_71_BM, the relatively higher portion of face-on orientation in **PTT-ODTTBT**:PC_71_BM film supports a higher PCE in the corresponding device, because this orientation would be more advantageous for charge transfer.

Furthermore, the enhanced device performance for both types of OPVs after adding DIO can be attributed to increased face-on molecular orientation. The surface morphologies of both blend films were investigated by tapping mode atomic force microscopy (AFM) and transmission electron microscopy (TEM). As shown in Figure 6, the root-mean-square (RMS) of the blend films without DIO was measured to be 1.79 and 3.87 nm for **PT-****ODTTBT**:PC_71_BM and **PTT-ODTTBT**:PC_71_BM, respectively. The relatively high RMS value of the latter was attributed to the high crystallinity of the polymer upon incorporating TT on its backbone, as shown in the GIWAX data. After adding DIO, both blend films displayed smooth surfaces with much lower RMS values (0.99 and 1.75 nm for **PT-ODTTBT** and **PTT-ODTTBT**, respectively). As shown in the TEM images (Figure 6e–h), the **PTT-ODTTBT**:PC_71_BM film exhibited a bicontinuous interpenetrating network with a well-developed fibrillar nanostructure, which could be beneficial for charge transport and separation. In comparison, phase separation between the donor and acceptor was observed in the **PT-ODTTBT**:PC_71_BM film, suggesting that the poor miscibility of this blend film would affect the charge transport and separation, leading to relatively lower *J*_sc_ and *FF* than those of **PTT-ODTTBT**:PC_71_BM.

## 4. Conclusions

2-Octyldodecyl-substituted thieno[3,2-*b*]thiophene was successfully synthesized as a new building block via cobalt (II)-catalyzed alkylation. Two corresponding conjugated copolymers (**PT-ODTTBT** and **PTT-ODTTBT**) composed of benzothiadiazole and thiophene or thieno[3,2-*b*]thiophene were synthesized as electron donor materials for organic photovoltaics. PTT-ODTTBT exhibited a strong intermolecular *π*–*π* stacking even in the solution state and a well-ordered lamellar stacking than **PT-ODTTBT**, leading to improved hole mobility. The PTT-ODTTBT:PC_71_BM-based OPV device exhibited the highest PCE of 7.05% and higher short-circuit current and fill factor than the **PT-ODTTBT**:PC_71_BM-based device. Such improved OPV device performance is attributed to the high crystallinity, preferential face-on orientation, and formation of a bicontinuous interpenetrating network in the **PTT-ODTTBT**:PC_71_BM film. Incorporation of thieno[3,2-*b*]thiophene in the polymer backbone would be a good approach for enhancing the molecular crystallinity and hole mobility in organic photovoltaics.

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
