# Peer review of "Novel Conjugated Polymers Containing 3-(2-Octyldodecyl)thieno[3,2-b]thiophene as a π-Bridge for Organic Photovoltaic Applications"

_polymers, 2020, doi:10.3390/polym12092121_

Round 1

Reviewer 1 Report

The authors report on a 3-(2-Octyldodecyl)thieno[3,2-b]thiophen synthesized as a new bridge with a long branched side alkyl chain. Two donor-p-bridge-acceptor type copolymers were designed and synthesized by combining this p-bridge structure, a fluorinated benzothiadiazole acceptor unit, and a thiophene or thienothiophene donor unit, (PT-ODTTBT or PTT-ODTTBT respectively) through Stille polymerization.The paper in general is quite novel, contains a great number of scientific analytical techniques and could be accepted for publication after including some minor comments.
1. Please include in abstract the active area of the OPV devices fabricated, the method for the device fabrication (e.g. spin coating or printing technique as i.e. slot-die printed OPV), the standard deviation of the max. PCE of 7.05%, the device architecture (normal or inverted stuck), the fabrication parameters e.g. in ambient atmosphere or N2 within a glovebox. All the above details improve significantly the paper and attract the reader's attention.
2. Please include in the introduction a paragraph with the highest efficiency OPV devices as state of the art existing PCE in literature: method for fabrication i.e. spin coating fabrication or with scalable roll-to-roll printing processes, the stucking of layers (normal or inverted stucks), as well as active area of the reported PCE values and the molecular architectures of the donor - acceptor materials.
3. Please include a paragraph with potential usage of the synthesised polymers in this study for large scale production/ printing of OPVs; e.g could the polymers in this study be processed/ deposited in ambient atmosphere (pls refer to printed OPV devices in literature Energy Environ. Sci., 2012,5, 5117-5132; Solar Energy Materials and Solar Cells, Volume 144, 2016, Pages 724-731; Nanoscale, 2010,2, 873-886; Mater Today Proceedings Volume 3, Issue 3, 2016, Pages 832-839).

The paper contains valuable information and several results to point out the innovative new synthesised donor-p-bridge-acceptor type copolymers by combining this p-bridge structure, a fluorinated benzothiadiazole acceptor unit, and a thiophene or thienothiophene donor unit, (PT-ODTTBT or PTT-ODTTBT respectively) through Stille polymerization.

Several things need further discussion as well as improvements in terms of representation improving especially the "short" and weak introduction section of this work.

In terms of originality, scientific quality, relevance & contribution to the field and presentation, this is a manuscript of good level.

The findings of the paper are sufficiently novel to warrant its publication, however, after including and considering all the minor changes proposed.

Author Response

Response to referees comments (Reviewer: 1)

Many thanks for your evaluation and valuable comments on our manuscript. We have answered the reviewer’s comments as best we can.

Q1) Please include in abstract the active area of the OPV devices fabricated, the method for the device fabrication (e.g. spin coating or printing technique as i.e. slot-die printed OPV), the standard deviation of the max. PCE of 7.05%, the device architecture (normal or inverted stuck), the fabrication parameters e.g. in ambient atmosphere or N2 within a glovebox. All the above details improve significantly the paper and attract the reader's attention. 

A1) We provided detailed information about the device fabrication process including the active area of the fabricated OPV devices in the abstract according to the referee’s comment.

Q2) Please include in the introduction a paragraph with the highest efficiency OPV devices as state of the art existing PCE in literature: method for fabrication i.e. spin coating fabrication or with scalable roll-to-roll printing processes, the stucking of layers (normal or inverted stucks), as well as active area of the reported PCE values and the molecular architectures of the donor - acceptor materials.

A2) We added the information about the stage of art in OPV devices including the fabrication methods in the Introduction part according to the referee’s comment.

Q3) Please include a paragraph with potential usage of the synthesised polymers in this study for large scale production/ printing of OPVs; e.g could the polymers in this study be processed/ deposited in ambient atmosphere (pls refer to printed OPV devices in literature Energy Environ. Sci., 2012,5, 5117-5132; Solar Energy Materials and Solar Cells, Volume 144, 2016, Pages 724-731; Nanoscale, 2010,2, 873-886; Mater Today Proceedings Volume 3, Issue 3, 2016, Pages 832-839).

A3) We cited the recommended references in the “reference 11-15” according to the referee’s comment. For the applicability of our polymers to large scale production, we haven’t get enough supporting data for practical application yet. We may need more supporting information and experiments to mention the possibility to real application of our product so we didn’t mention about that in the revised manuscript. We expect referee’s understanding for this.

Reviewer 2 Report

In this work, two donor copolymers named PT-ODTTBT and PTT-ODTTBT were designed and synthesized. OPVs based on PTT-ODTTBT/PC71BM exhibited PCE of 7%. The topic of new donor polymers, especially medium bandgap donor polymers, is very important recently since their complementary absorption range with new narrow bandgap non-fullerene acceptors. In my opinion, this work can be accepted after minor revisions. Some technical comments are shared:

  1. In the Introduction part, a recent review paper (Acc. Chem. Res., 2020, 53, 1218–1228) on OPV field is recommended.
  2. In Fig.1, there seems minor absorption shift between solution sample and film sample which indicate small aggregation tendency. However, the planarity of these new polymers looks good, more discussion is appreciated.
  3. Considering the bandgaps and energy levels, these polymers exhibit great potential to match with narrow bandgap non-fullerene acceptors. Have the authors made some devices based on new polymer/non-fullerene acceptor?
  4. In Fig.6, the scale bar of different height is missing.

Author Response

Response to referees comments (Reviewer: 2)

Many thanks for your evaluation and valuable comments on our manuscript. We have answered the reviewer’s comments as best we can.

Q1) In the Introduction part, a recent review paper (Acc. Chem. Res., 2020, 53, 1218–1228) on OPV field is recommended.

A1) We cited the paper in the list of references (reference 3) according to the referee’s comment.

Q2) In Fig.1, there seems minor absorption shift between solution sample and film sample which indicate small aggregation tendency. However, the planarity of these new polymers looks good, more discussion is appreciated.

A2) In this work, we design and synthesized new D-p-A type copolymers which showed good molecular planarity due to introduction of TT as a p-bridge. It is believed that the synthesized polymers have a strong aggregation tendency due to their high molecular planarity, and thus showed an absorption spectrum similar to that of a film state even in a solution state. On the film states, synthesized polymers showed distinct vibronic absorption band at around 680 – 690 nm, which means strong pi-pi stacking and aggregation tendency. We discussed about that in the manuscript according to the referee’s comment.

Q3) Considering the bandgaps and energy levels, these polymers exhibit great potential to match with narrow bandgap non-fullerene acceptors. Have the authors made some devices based on new polymer/non-fullerene acceptor?

A3) As referee pointed out, the synthesized polymers could show good performances with low band gap non-fullerene acceptors which have complimentary absorption to the our donor polymers. The polymers were synthesized a few years ago and the student who made them left lab so we couldn’t expand the work with non-fullerene acceptors yet. We plan to do some work with the non-fullerene acceptors in the near future.

Q4) In Fig.6, the scale bar of different height is missing.

A4) We corrected the scale bar of AFM images in Fig. 6 according to the referee’s comment.